# Open Earth Observations for Sustainable Urban Development

**Mihir Prakash** [1,*], **Steven Ramage** [2], **Argyro Kavvada** [3,4]  **and Seth Goodman** [1]

1 Global Research Institute, College of William and Mary, Williamsburg, VA 23185, USA; sgoodman@aiddata.wm.edu
2 Group on Earth Observations Secretariat, 2300 Geneva, Switzerland; sramage@geosec.org
3 National Aeronautics and Space Administration, Earth Science Division, 300 E St. SW, Washington, DC 20546, USA; argyro.kavvada@nasa.gov
4 Booz Allen Hamilton, 8283 Greensboro Dr, McLean, VA 22102, USA
* Correspondence: mprakash@aiddata.wm.edu or mprakash@aiddata.org

**Abstract:** Our cities are the frontier where the battle to achieve the global sustainable development agenda over the next decade would be won or lost. This requires an evidence-based approach to local decision-making and resource allocation, which can only be possible if current gaps in urban data are bridged. Earth observation (EO) offers opportunities to provide timely, spatially disaggregated information that supports this need. Spatially disaggregated information, which is also demanded by cities for forward planning and land management, has not received much attention largely due to three reasons: (i) the cost of generating this data through traditional methods remains high; (ii) the technical capacity in geospatial sciences in many countries is low due to a shortage of skilled professionals who can find and/or process available data; and (iii) the inertia against disturbing routine workflows and adopting new practices that are not imposed through legal requirements at the country level. In support of overcoming the first two challenges, this paper discusses the importance of EO data in the urban context, how it is already being used by some city leaders for decision making, and what other applications it offers in the realm of urban sustainability monitoring. It also illustrates how the EO community, via the Group on Earth Observations (GEO) and its members, is working to make this data more easily accessible and lower barriers of use by policymakers and urban practitioners that are interested in implementing and tracking sustainable development in their jurisdictions. The paper concludes by shining a light on the challenges that remain to be overcome for better adoption of EO data for urban decision making through better communication between the two groups, to enable a more effective alignment of the produced data with the users' needs.

**Keywords:** sustainable cities; urban data; Earth observation; new urban agenda; sustainable development goals; data for development

## 1. Introduction

Cities are where international development and environmental policy agreements (Box 1) can have the greatest impact. They are the lifeblood of the engines that drive economic growth and have the potential to help realize the goal of sustainable development. According to the United Nations (UN), nearly 70% of the world's population will live in cities by 2050 [1]. This puts cities at the forefront not only for delivery of services but also in tackling the adversities of climate change through mitigation and adaptation measures. They are places where nature-based solutions and ecological or sustainable thinking can materialize through better urban design and planning.

Sustainable urbanization needs action at all levels of administration—local, provincial, and national. This requires sufficient, spatially disaggregated data as the foundation upon which action towards national sustainability would need to be built. Resource constraints, coupled with only a 10-year runway remaining to achieve the Sustainable Development Goals (SDGs) necessitate strategic, evidence-based decisions that bring us closer to sustainability by 2030. As the movement among mayors and other local leaders to align their development with sustainability principles gathers momentum, the demand for cities' data is experiencing growth [2]. However, the supply is lagging. Increasingly, it is being recognized that Earth observation (EO) data can complement or enhance traditional data sources for cities and urban areas [3].

**Box 1.** Which international agreements are relevant to cities?

**Sustainable Development Goals (SDGs)**—adopted by all UN Member States in 2015 as a universal call to action to end poverty, protect the planet and ensure that all people enjoy peace and prosperity by 2030.
**New Urban Agenda (NUA)**—adopted at the Habitat III conference in 2016, it is a shared vision for a better and more sustainable future, one in which all people have equal rights and access to the benefits and opportunities that cities can offer, and in which the international community reconsiders the urban systems and physical form of our urban spaces to achieve this.
**Sendai Framework**—endorsed by the UN General Assembly after the 2015 3rd World Conference on Disaster Risk Reduction, the framework is a call to action for member states to reduce disaster risks in collaboration with local governments and other relevant stakeholders.
**Paris Agreement**—is a 2016 agreement within the UN Framework Convention on Climate Change (UNFCCC) dealing with greenhouse gas emissions reduction and finance.

To be able to generate timely, reliable and disaggregated data on the 232 indicators in the SDG framework, PARIS21 finds the need to double the financing for data and statistics to ensure that the "SDGs leave no-one behind" [4]. This estimate is based on the data needs for reporting on national SDG progress. Data needs of cities for forward planning, monitoring local development and reformulating local policies imply additional financing needs beyond those estimated by PARIS21.

Urban planning and land management data does not typically come from national sources. These are data that cities and municipalities often generate themselves—either in real-time, or regularly, or as needed. Some examples of such data generating activities are cadastral surveys, land use surveys, topographical surveys, transportation and traffic surveys, air quality monitoring, and ground and surface water monitoring. To complement this information, national statistical systems such as censuses and sample surveys, provide urban demographic and socioeconomic data. Other relevant national sources of data include meteorological departments, geography, and geology department surveys.

The 2030 Agenda for Sustainable Development, or the SDGs, recognizes the need for indicators to be disaggregated, where relevant, by income, sex, age, race, ethnicity, migratory status, disability, and geographic location [5]. Without this geographic disaggregation, information is of little policy relevance beyond monitoring the SDGs at a high-level. Geographic disaggregation was included in the agenda to provide knowledge of hotspots where policy interventions are required to make inclusive progress towards the globally agreed goals and targets.

However, geographically disaggregated (or geospatial) data has not received much attention and remains a challenge to find. This is also data that is used by cities for forward planning and land management. There are three key reasons for this: (i) the cost of generating this data through traditional methods remains high; (ii) the technical capacity in geospatial sciences in many countries is low due to a shortage of skilled professionals who can find and/or process available data; and (iii) the inertia against disturbing routine workflows and adopt new practices that are not imposed through legal requirements at the country level. Earth observation (EO) data and engagement with the global geospatial community can help overcome the first two challenges. The third challenge requires continued advocacy efforts to build the necessary political will. In support of overcoming the first two challenges, this paper discusses the importance of EO data in the urban context, how it is being used by city leaders for decision making already, what other applications it offers in the realm of urban

sustainability monitoring, and what is being done in the EO community to lower barriers of use of this data by policymakers and urban practitioners that are interested in implementing and tracking sustainable development in their jurisdictions. In doing so, this paper also aims to communicate the needs of urban data users to the EO data producers, and the offerings of EO data producers to the urban data users, and also shine a light on the challenges that need to be overcome for better adoption of EO data for urban decision making through better communication between the two (Figure 1).

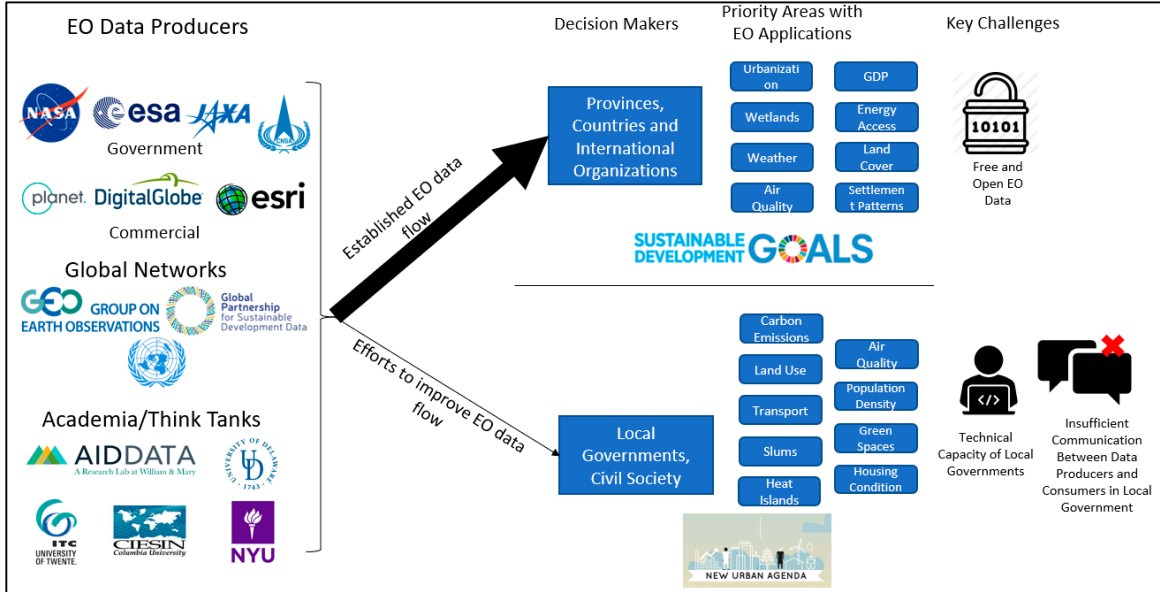

**Figure 1.** Earth observation (EO) application areas in decision making, strength of existing data flows, and challenges that need to be overcome. Note: the list of data producers, areas of EO applications and key challenges above are not exhaustive and are used for illustrative purposes only. Issues such as data complexity, data storage, management, etc., also require attention to further improve established EO data flows.

## 2. Earth Observations: How Can This Data be Used by City Leaders?

Earth observations are data and information about the planet's physical, chemical, and biological systems [6]. It involves monitoring and assessing the status of, and changes in, the natural and man-made environment. In recent years, Earth observations have become more sophisticated with the development of remote-sensing satellites and increasingly high-tech in situ instruments that measure on the ground data. Today's Earth observation instruments include floating buoys for monitoring ocean currents, temperature, and salinity; land stations that record air quality and rainwater trends; seismic and global positioning system (GPS) stations; drones for capturing high resolution images in small scales; and high-tech environmental satellites that scan the Earth from space.

EO is more important than ever now due to the impacts of human civilization on the global environment. EO data, whether atmospheric, oceanic, or terrestrial, are already helping decision makers around the world to better understand the issues they face, and shape more effective policies aligned with local goals. EO data can be used to support national statisticians, chief data scientists, chief data officers, ministers of planning, or others concerned with evidence in support of sustainable development [7].

With advances in space-based instruments, EO capabilities have also seen a substantial increase in the range of information they can provide for decision-making. Tracking atmospheric conditions, measurements of vegetation and land use/land cover, estimating population densities and the scale of electrification are just a few drops in the ocean of applications. However, the greatest value proposition of EO data is its continuous spatial-temporal coverage at a fraction of the cost of traditional methods, while ensuring objectiveness, comparability as well as sustainability of services. This value proposition

led world leaders to acknowledge the important role that EO has to play in making the SDG framework "feasible through the provision of essential evidence, including the tracking of indicators over time, and supporting the implementation of solutions to reach specific targets" [8].

## 2.1. Using EO Data for City Planning and Management

Like the value identified by national leaders, EO also poses great opportunities for leaders at the local level. Land use mapping, topographic mapping, and other useful information for cities can now be generated using high-resolution (or images that capture smaller objects on the ground) satellite imagery at low cost. High-resolution imagery, when analyzed, provides sufficiently detailed information and reliable data can be generated about areas as small as a single house. For example, the city of Kandy in Sri Lanka, with the support of the World Bank, developed various land use maps to take stock of its current land consumption and land use activities using EO at a fraction of the cost and time taken by traditional survey methods (Figure 2).

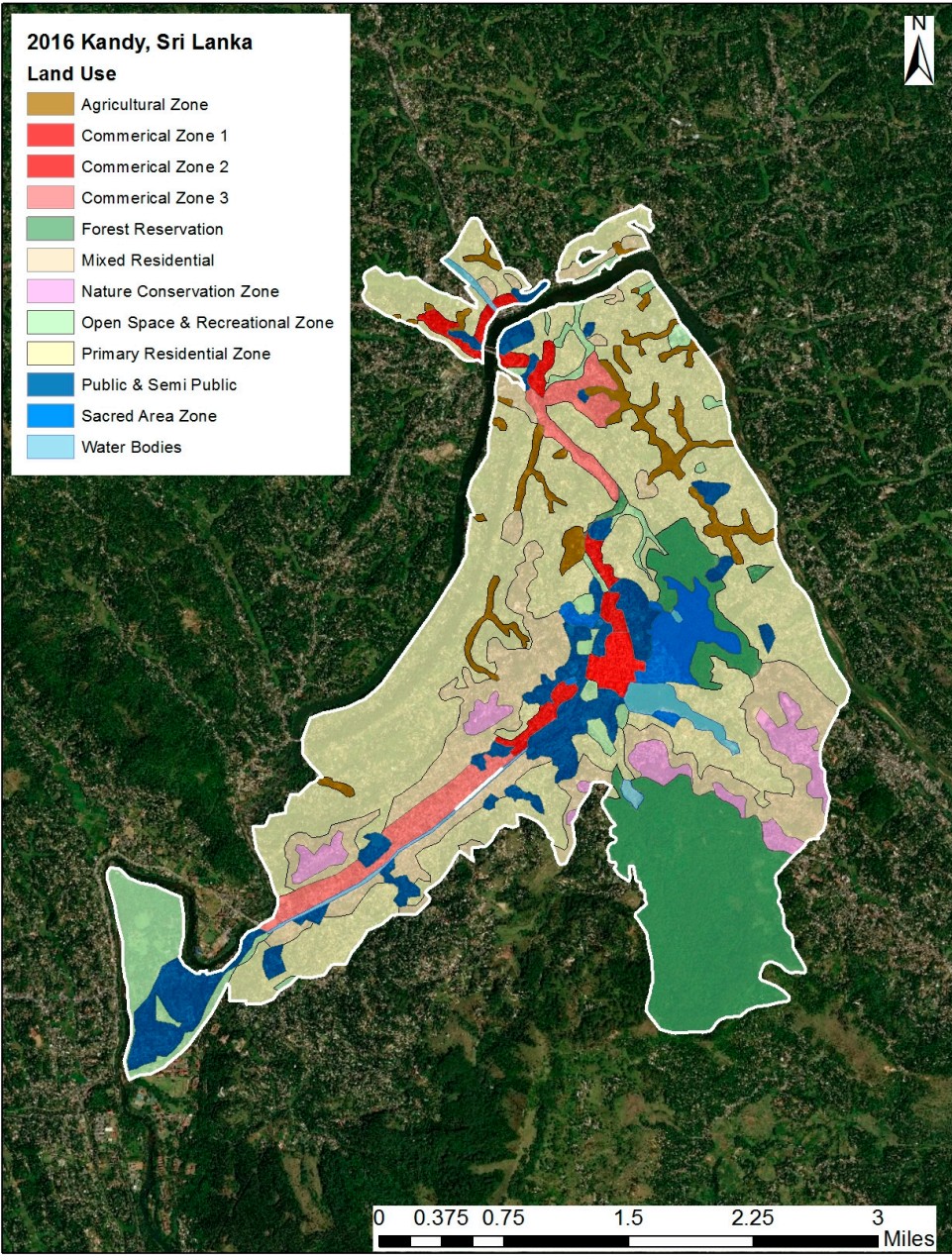

**Figure 2.** Land use map of Kandy, Sri Lanka in 2016 drawn using remote sensing [9].

Urban monitoring need not just rely on the use of high-resolution satellite imagery. Medium-low resolution images (15–30 m) also offer valuable insights that are useful for urban managers and decision makers. For example, Kampala, Uganda used medium-low resolution imagery to monitor the overall growth and direction of sprawl of its built-up areas (Figure 3). Information on land use change and spatial growth rates and direction can inform city leaders on areas that require administrative attention, for example, to regulate construction and avoid encroachments on public land, channel resources to create supporting infrastructure, estimate demand for services, and conserve ecologically sensitive areas. The European Space Agency's Copernicus program is working on providing similar built-up area maps for all cities worldwide with at least biannual updates starting in the year 2021. The program will also release the World Settlement Footprints dataset, using Sentinel-1 and Landsat data that will have information on the urban footprint of all settlements in 2015. The World Settlement Footprints dataset will be available in summer 2020 [10].

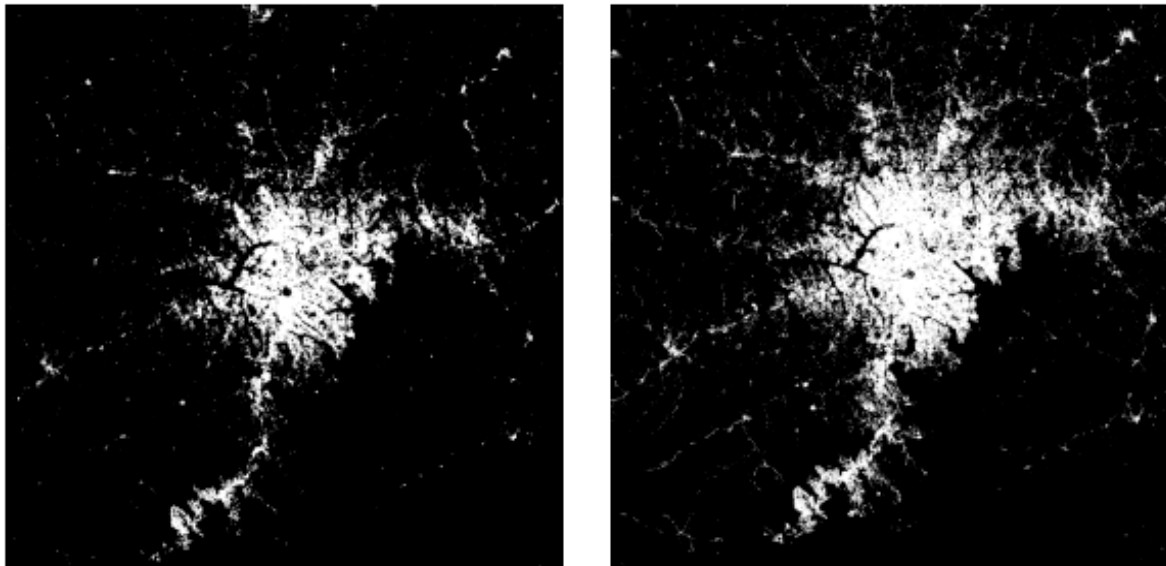

**Figure 3.** Urban extent changes in Kampala, Uganda between 2002 and 2014 [8].

EO also has applications for gathering data on environmental attributes of cities. For example, urban planners and environmental scientists continue to use remotely sensed information as a way to measure land surface temperatures at scale to inform policies on target areas within the city that require mitigation measures, such as urban greening, to tackle challenges associated with the heat island effect (Figure 4). Surface temperature readings can be combined with other measures of weather such as precipitation, soil moisture, and surface water quality or quantity over time to develop predictive models that allow for future-proofing urban development. This is particularly relevant for cities grappling with the effects of climate change, such as sea level rise and urban flooding.

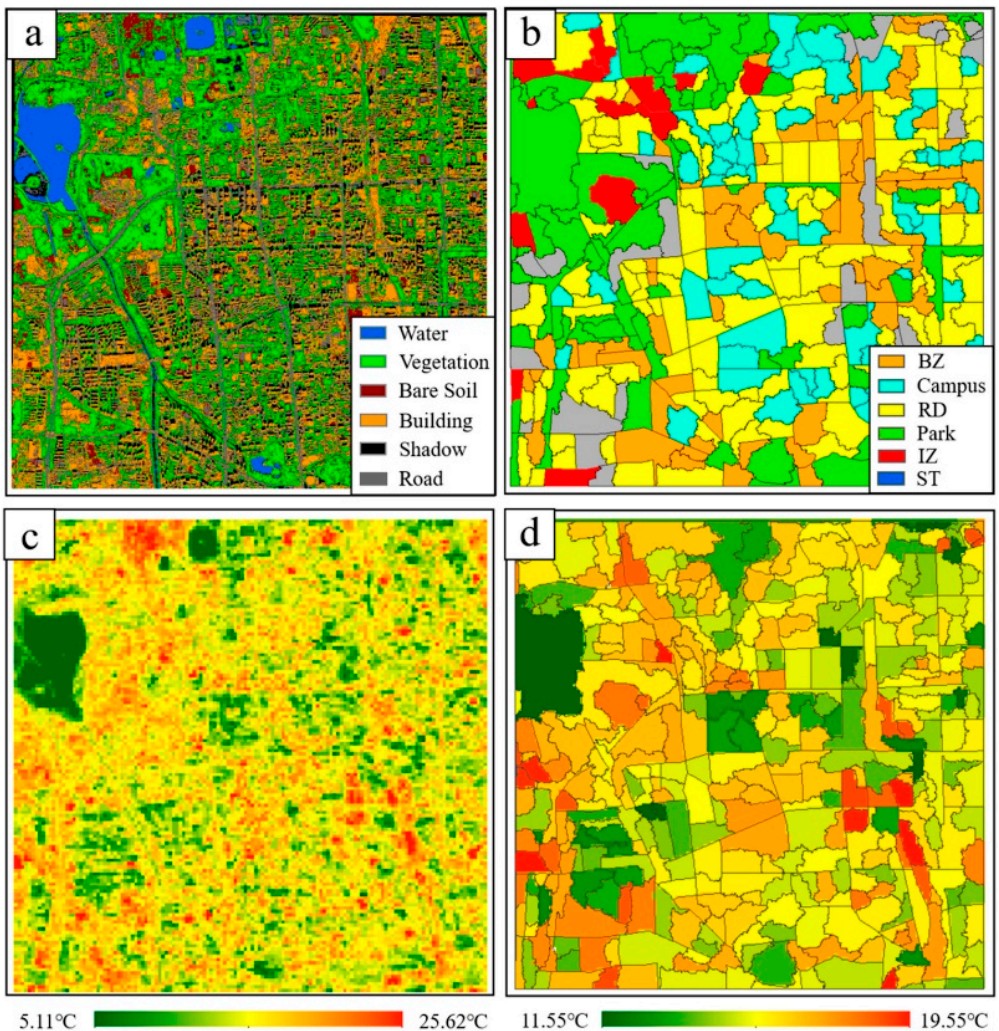

**Figure 4.** Urban heat island mapping in Beijing, China [11].

## 2.2. Using EO Data to Track and Report on International Agreements at the City Level

City leaders that adopt the international frameworks of sustainable development in their jurisdictions can use EO to track and report progress on many of the global indicators. For example, indicator 11.6.2 of the SDG framework that tracks annual mean levels of fine particulate matter in cities, also referenced in paragraphs 13, 55, and 67 of the NUA, has traditionally been monitored through ground-based measurement instruments, which many cities in developing countries may or may not have. Earth observation can unambiguously assist in this measurement to be done at scale (Figure 5).

EO can also help in tracking other SDG indicators that are relevant to city-level spatial planning. Two such examples are SDG indicator 11.3.1: ratio of land consumption to population growth rate (Figure 6) and SDG indicator 11.2.1: proportion of population that has convenient access to public transport (Figure 7). For the former, EO can provide timely measures of the physical boundary of cities (built up area) that can be combined with population growth data from other sources [12]. For the latter, EO can provide information on where residential areas are and their density, which can be combined with information on transit routes to measure access.

An intrinsic value proposition of EO is the standardized approach it takes for measurements. This means that using the same methodology leads to results in different areas that are comparable. Issues arising from inconsistencies of definitions between two or more locations or human error, as could be in the case with traditional approach to statistics, do not affect results from EO. For example,

using the methodology from Bailey et al. (2019) [13], the SMURBS/ERA-PLANET H2020 project provides comparable and standardized air quality readings for the entire continent of Europe (Figure 5).

This standardized approach can prove advantageous for decision makers that are looking for ways to take objective stock of a situation. For example, the identification and mapping of slums over large areas can easily be done using a combination of EO and machine learning at low-cost (Figure 8).

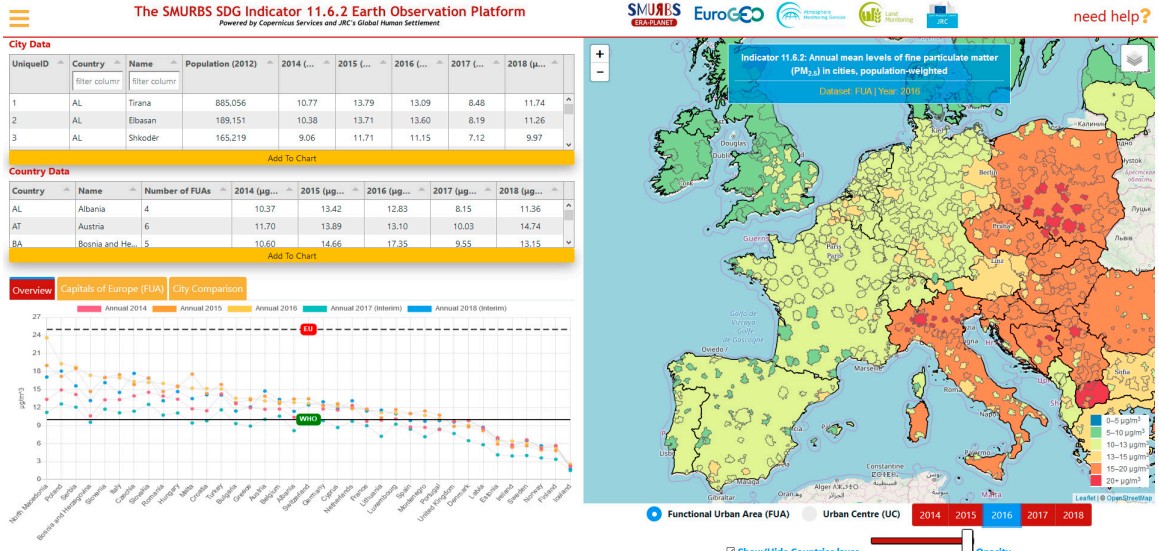

**Figure 5.** The online platform created by the SMURBS/ERA-PLANET H2020 project for monitoring the Sustainable Development Goal (SDG) indicator 11.6.2 (mean annual levels of fine particulate matter in cities, population-weighted) in Europe.

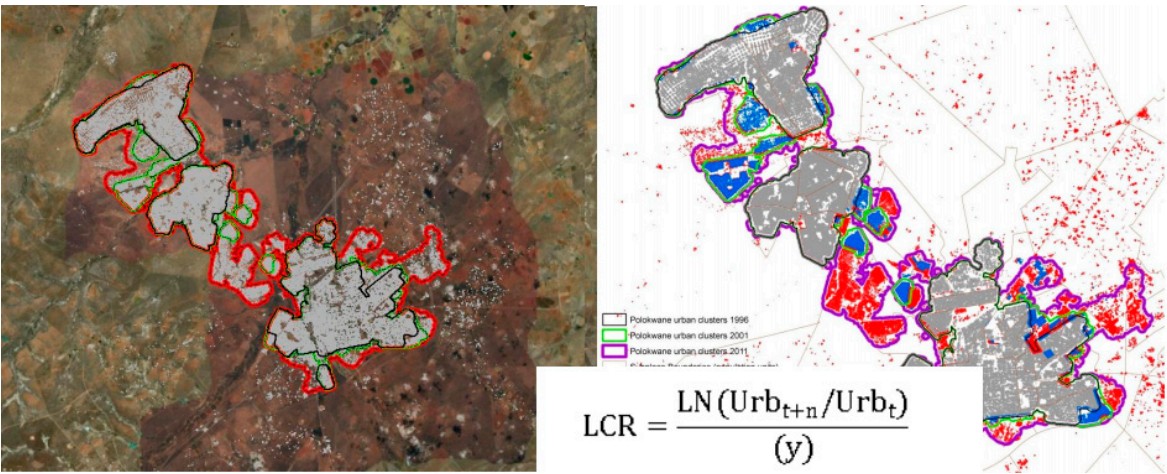

$$LCR = \frac{LN(Urb_{t+n}/Urb_t)}{(y)}$$

**Figure 6.** Measuring the land consumption rate [14].

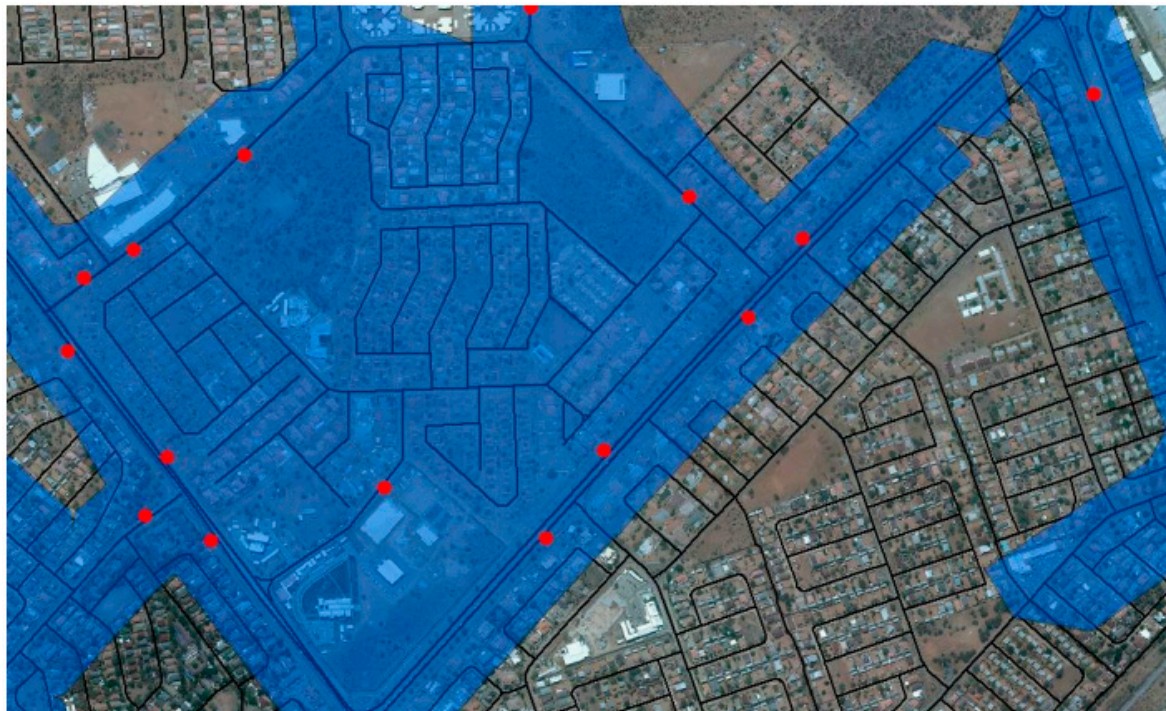

**Figure 7.** Evaluating access to public transport [14].

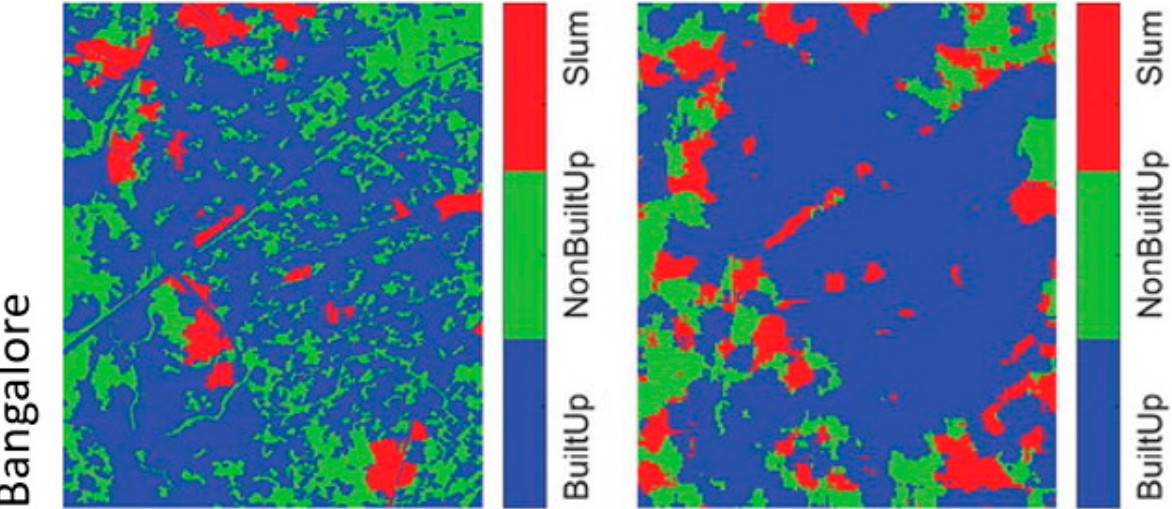

**Figure 8.** Identifying slum areas in Bangalore, India using remote sensing (left—identified slum areas using ground observations and right—identified slum areas using remote sensing) [15].

Table 1 provides some example cases of issues related to urban development that can be monitored using EO, and how they map to various international development agendas. This list is not exhaustive and is shared for illustrative purposes only. Where possible, specific goals and targets of these international agendas have been included. Check marks suggest relevance to the global framework but not an exact target/indicator match.

**Table 1.** Applications of EO to various dimensions of global agendas.

| Urban Development Issues that Can Be Measured Using Earth Observation | SDGs | NUA | Sendai Framework | Paris Agreement | Relevant Sources |
|---|---|---|---|---|---|
| **Land Use** | | | | | |
| Land Consumption | 11 | | | | Atlas of Urban Expansion; Trends.Earth; Urban Mapper |
| Public Space/Recreational Use/Open Spaces | 11 | ✓ | | | ESA Sentinel-2 MSI; MODIS Land Cover |
| Length of Roads/Area Under Paved Roads | 11 | ✓ | | | Global Urban Observatory |
| **Housing** | | | | | |
| Population Density | 11 | ✓ | | | Gridded Population; Global Human Settlements Layer |
| Population Living in Slums | 11 | ✓ | | | Global Urban Observatory; www.urban-tep.edu |
| Condition of Dwelling (roof) | 11 | ✓ | ✓ | | www.urban-tep.edu |
| **Transportation** | | | | | |
| Access to Public Transportation | 11 | ✓ | | | Global Urban Observatory |
| **Water** | | | | | |
| Soil Moisture Content | 6, 15 | | | | NASA |
| Surface Water | 6 | ✓ | | | ESA; MODIS; www.SDG661.app |
| Wetlands Monitoring | 15 | ✓ | | | NASA |
| Precipitation | 6, 11 | ✓ | | | University of Delaware |
| **Air Quality** | | | | | |
| PM 2.5/10 | 3, 11 | ✓ | | | SMURBS |
| Temperature/Heat Islands | | ✓ | | | University of Delaware; EXTREMA Project |
| **Energy** | | | | | |
| Energy Access/Electrification | 7, 11 | ✓ | | ✓ | NOAA Nighttime Lights (VIIRS, DMSP) |
| **Climate Change** | | | | | |
| Atmospheric Carbon Concentration | 13 | ✓ | | ✓ | NASA JPL |
| Drought/Flood Monitoring | 11, 13 | ✓ | ✓ | | www.SDG661.app |
| Surface Temperature | 11 | ✓ | ✓ | | MODIS (MOD11) |
| **Other** | | | | | |
| Gross Domestic Product | 8, 11 | ✓ | C1,D1 | ✓ | NOAA Nighttime Lights (VIIRS, DMSP) |
| Critical Infrastructure | | | D1 | | NOAA Nighttime Lights (VIIRS, DMSP) |

Despite the low-cost nature of using open Earth observations, advances in EO and the opportunities it offers to the urban data ecosystem the EO community have been unable to trigger its large-scale adoption and use. This is often because of the technical capacity of local government offices and, possibly, the inertia against integrating EO data into everyday activities that changes the operational status quo. The next section discusses what the EO community is doing to make EO data more accessible to municipal decision makers.

## 3. Earth Observations: Overcoming Capacity Challenges in Cities

Earth observation is a specialized science with a high entry barrier. Trained professionals with satellite image processing capabilities are limited in number, as compared to the need. Remote sensing jobs tend to be concentrated in specific institutions that specialize in remote sensing without contextual knowledge related to international development. The cohort of remote sensing specialists who work on urban studies is a further niche. Therefore, local government capacity to use remotely sensed data and geographical information system (GIS) software would need to be developed over the next decade to fully utilize the opportunities EO offers to cities with the ability to contextualize analyses to local needs.

Efforts are underway globally in advocating for governments at all levels to build such technological capabilities through strategic investments into human resource and computing infrastructure. Founded in 2005 at the 3rd Earth Observation Summit, the Group on Earth Observations (GEO) is an intergovernmental partnership that has been advocating for the use of openly available Earth observation data to tackle issues related to sustainable development. Working together with 109 Member Governments, GEO has made available over 400 million freely available EO data resources on the global Earth observation system of systems (GGEOSS) portal. In addition, GEO is supporting

national level policy makers to make use of EO data in their decision-making through the development of the GEO Knowledge Hub, which is an open-source digital library designed to serve the community and to ensure that all these components are visible, accessible, and that others can share their knowledge and experience in a codified manner for country-relevant, evidence-based decision-making in order to allow this knowledge to be reproduced. It will allow one to advance reproducibility of the knowledge produced by the GEO community [16].

### 3.1. The Group on Earth Observations: Supporting Sustainable Cities with Coordinated Earth Observations

GEO coordinates a global work program of over 50 activities, using open EO, to address environmental and social challenges, many of which are keenly applicable to the challenges facing cities. Some of GEO's most notable initiatives of relevance to sustainable urban development are the Global Urban Observation and Information (GUOI) initiative, the GEO Human Planet Initiative (HPI), and the Earth observation in support of the Sustainable Development Goals (EO4SDG) initiative.

GEO's GUOI is improving urban monitoring and assessment through international cooperation and collaboration to provide datasets, information, and technologies to pertinent urban development professionals operating at the national and international level. Government agencies that typically use these EO datasets include departments of urban and regional planning, environmental management, natural resources, metropolitan transit authority, and office of sustainability, and regional statistics. GEO supports these agencies through technical assistance with accessing and developing robust datasets on urban land use/land cover, urban form and growth patterns, infrastructure and transport needs, ecosystems and biodiversity, human health, thermal comfort, food security, and socioeconomic development.

GEO's HPI provides services to international institutions and other stakeholders with next generation measurements and information products that provide new scientific evidence on humanity's impacts on the planet. A recent outcome of the HPI is the new 'Degree of Urbanization' dataset, which provides a universal and harmonized definition of cities based on population density and settlement size [17]. This project was cocreated by the European Commission, the Organization for Economic Cooperation and Development (OECD), Food and Agricultural Organization (FAO), UN-Habitat, and the World Bank, and its outcomes were under consideration for adoption in March 2020 by the UN Statistics Commission.

GEO's EO4SDG initiative works closely with SDG custodian agencies responsible for specific indicators relevant to their thematic expertise and mandate—such as the UN Environment Program, the UN Convention to Combat Desertification (UNCCD), and the UN Habitat—on SDG indicator method development, testing, refinement, adoption, and widespread, sustained use. These efforts have led to proposals within the UN system to elevate the readiness status of global methodologies related to fresh water, terrestrial ecosystems, and sustainable urbanization, with Earth observations integrated as notable inputs. As an example, UN Environment has partnered with Google, the European Commission's Joint Research Centre (JRC), the European Space Agency (ESA), the United States National Aeronautics and Space Administration (NASA), and GEO to provide free and open EO data in a way that is policy-friendly, so that governments and relevant stakeholders can easily assess and visualize data on surface water extent (sdg661.app). EO4SDG is working with UN Environment to identify additional global EO datasets that can be incorporated in this application to support official global to regional and national level SDG monitoring and reporting. In 2020, UN Environment plans to incorporate the global mangrove watch (GMW) dataset as an additional dataset for indicator 6.6.1 (water-related ecosystems). The GMW utilizes ALOS-2 PALSAR-2 data to detect changes in the forest extent. Recent efforts led by NASA in collaboration with Japan's Aerospace Exploration Agency (JAXA) and other contributors have focused on updating global mangrove extent baselines using available data in an automated way, applicable at global scale. This is particularly key given current and upcoming radar (NISAR, ALOS 4) and optical satellites (Sentinel-2, Landsat 8/9) planned by a range of international space agencies.

In partnership with UN-Habitat and as part of the GEO EO4SDG Initiative, Conservation International (CI) and NASA have collaborated to scale the use of EO data to support the monitoring and reporting of SDG indicator 11.3.1 (land use efficiency) for cities across the globe (see: Box 2). EO data allow for the estimation of a city's built-up area extent as well as changes in urban-land cover over time. For this effort, a module in Trends.Earth—an online platform that uses cloud computing to process satellite images into usable information, assessing land trends—has been developed by CI in partnership with NASA (http://trends.earth/). In July 2019, NASA's Applied Sciences Remote Sensing Training Program (ARSET), CI, and UN Habitat conducted an advanced webinar series for local, regional, state, federal, and international organizations interested in using Earth observation data for SDG 11.3.1 monitoring and reporting [18,19].

**Box 2.** Earth Observations for Sustainable Cities and Communities Toolkit.

EO4SDG, in partnership with UN Habitat, HPI, GUOI, the GEO Secretariat, and partners, is in the process of developing a customizable and continually updated toolkit on the integration of Earth observation and geospatial information into the urban monitoring and reporting processes. The toolkit, developed through a consultative process with countries and cities, will complement guidance published by UN-Habitat [20], and will be produced in conjunction with UN-Habitat. It will define needs, data requirements, and provide practical guidance on the integration of remotely sensed and ground-based EO data with national statistics, socioeconomic data, and other, ancillary information to help countries monitor, report, and drive progress on SDG 11 and the NUA. The toolkit will facilitate knowledge sharing, connect national, subnational, and city experiences, and foster understanding between technical analysts and decision makers regarding the role and potential contributions of EO in support of SDG 11 and the NUA. GEO Members, and interested cities from member countries, are invited to contribute to the design and implementation of this toolkit, in accordance with their needs and priorities. More information is available via the EO4SDG web portal.

Besides these three work streams, other programs within GEO also have relevance to sustainable urban development. Few examples are: the resilience brokers initiative that works with stakeholders in the private sector to ensure that EO data is used effectively in disaster and emergency contexts; the SMURBS/ERA-PLANET H2020 project that promotes the "smart-city" concept through the integration of EO and other data at the urban level; the GEO biodiversity observation network (GEOBON) that tracks key biodiversity indicators and the Earth observation for city biodiversity index (EO4CBI) that extends this work to the city level; and GEO Wetlands that monitors wetland quality.

Despite many successes through these programs and beyond, GEO still has much work to do on bringing more higher resolution, proprietary data into the open data, open access realm. Many of the EO applications discussed earlier that involve identifying distinct objects that are only a few meters wide and long, such as identifying slum dwellings and measuring length of and access to roads, require high resolution imagery. Right now, most open source satellite images, such as those from Modis, Landsat, and Sentinel satellites, are medium–low resolution images (10 meters and coarser). As technology improves and providers of high-resolution imagery lower their barriers through the efforts of global partnerships, such as GEO, the use of EO data in urban analytics would likely see an increase. For example, Maxar Technologies and Ecopia.ai are extracting every building and road throughout 51 countries in sub-Saharan Africa using very high-resolution satellite imagery and machine learning [21]. As governments and private stakeholders share more of their datasets, such as recent initiatives by China and Japan, data that are more granular will become available on the world's cities [22,23].

### 3.2. GeoQuery and Other Geospatial Tools: Making EO Data Accessible

While GEO and others work on building political will and developing geospatial capacity at the city level, AidData, a research lab at William and Mary, has been working on expanding the functionalities of its GeoQuery tool to the city level (see: Box 3). GeoQuery is a public-good tool that allows users to start using data generated through EO by providing useful data in a spreadsheet format. The goal of this web-based tool is to reduce the technical barriers associated with utilizing spatial data

for policy planning and decision-making. GeoQuery is publicly available and powered by William and Mary's high-performance computing cluster, which automates the processing of terabytes of spatial data so that users are not limited by computational resources or technical skills. For example, users can request measures of air quality for their cities through an online form and receive an output emailed to them in spreadsheet format, which could then be visualized or analyzed on traditional spreadsheet software or mapped on GIS software, depending on the user's technical capability (Figure 9).

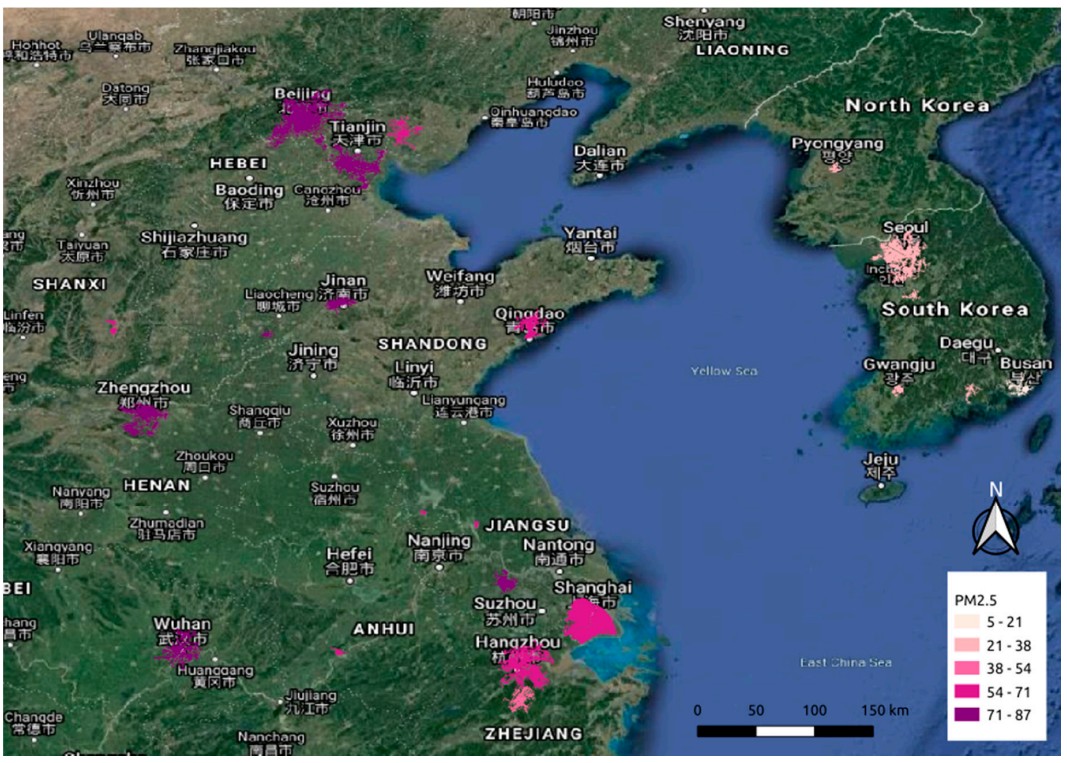

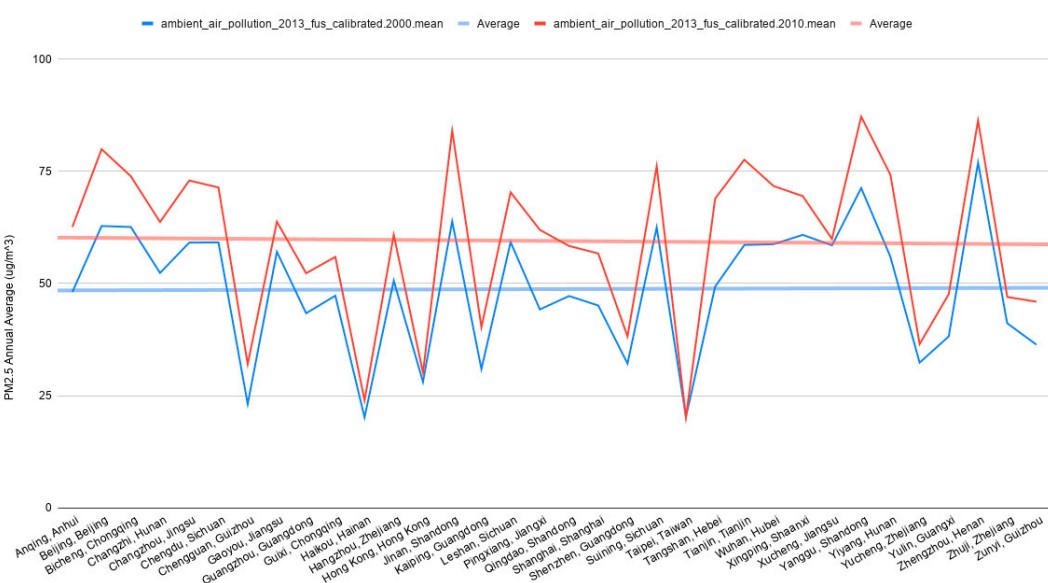

**Figure 9.** GeoQuery Output on air quality in Chinese cities mapped (**top**) and graphed (**bottom**) for 2000 and 2010.

GeoQuery is not the only tool that eliminates the need for skills in remote sensing. Other similar tools also exist, which provide information in forms that are usable by specific target audiences.

For example: PRIO Grid, created by the Peace Research Institute Oslo, offers a range of data aggregated to a global grid; the IPUMS Terra offers information on population and environmental data; Trends.Earth is a plugin for the open source desktop GIS software Quantum GIS that allows users to visualize land cover change over time; and MapX is an online platform that was developed by the UNEP that allows users to perform various GIS analyses and visualize data on natural resources. The overarching goal of these tools, and others like them, is to simplify working with geospatial data in varying contexts. However, GeoQuery is unique because it is not limited by its scope in terms of the datasets, boundaries, or applications for which is can be used. Another strong value proposition of GeoQuery is that it provides data that is has been aggregated to administrative/political boundaries, which makes it more fit for local decision-making purposes. However, being an open-source tool, GeoQuery is limited to sharing processed EO data that is freely available only. This means that some information is older while others are recent. As newer datasets become freely available, GeoQuery will update its data offerings and temporal coverage accordingly.

Nonetheless, any EO tool, including GeoQuery, can only generate data for those cities and other administrative units that have publicly available administrative/municipal boundaries. Many countries in the developing world are yet to create formal cadastral maps that delineate administrative boundaries of cities and other settlements, or even formally publish higher order administrative boundaries. In other cases, cities are not limited to their administrative boundaries. For instance, the greater Accra metropolitan area has twelve municipalities, but the city of Accra is only one of the twelve. This is one of the drivers of urban research into universal definitions and delineations of cities (such as the degree of urbanization project discussed earlier), including definitions based on population densities and built-up areas, which can be very different from actual administrative jurisdictions that elected officials actually represent (Figure 10).

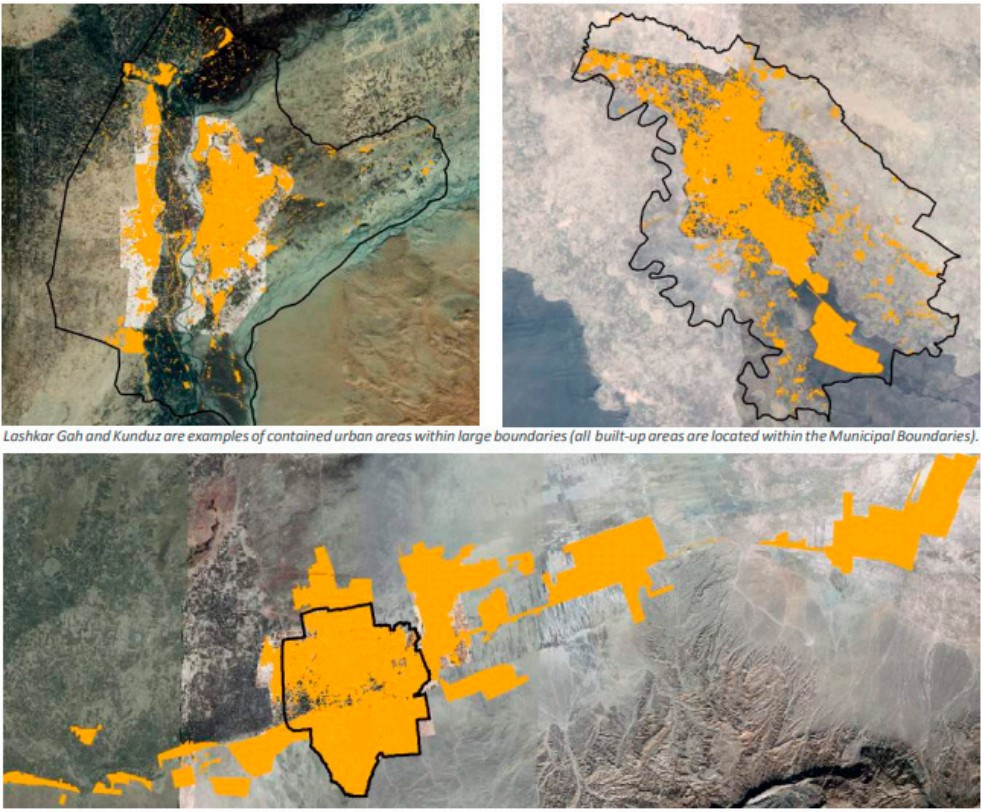

*Lashkar Gah and Kunduz are examples of contained urban areas within large boundaries (all built-up areas are located within the Municipal Boundaries).*

*Mazar-i-Sharif (above) and Ghazni (below) are examples of municipal boundaries that are too small because they do not accommodate all the built-up urban area. (Built-up areas within and outside the Municipal Boundary).*

**Figure 10.** Municipal boundaries versus built up area boundaries (yellow denotes built-up areas and black lines represent the municipal boundary) [24].

**Box 3.** GEOQUERY—What GEOQUERY is and how can it support cities?

The core repository of data for GeoQuery combines a curated collection of over 60 disaggregated spatial datasets with hundreds of administrative and other boundaries worldwide [25]. Users can browse this collection based on their desired area of interest, selecting boundaries which include administrative units as fine as administrative level 5 (ADM5) from GeoBoundaries, as well as a collection of 200 city boundaries from the Atlas of Urban Expansion. Based on the selected boundary and the time-period of interest, custom datasets are then created per the user's request and emailed for easy download and use.

**Example datasets available through GeoQuery**

| Theme | Dataset | Source |
|---|---|---|
| Socioeconomic | Nighttime lights (DMSP) | Nighttime lights (DMSP) |
| | Nighttime light (VIIRS) | Elvidge et al. 2017 |
| | Travel time to major cities | Nelson 2008 |
| | Population (Count and Density) | CIESIN 2000 |
| | Conflict event counts | Raleigh et al. 2010 |
| | Conflict deaths | Sundberg and Melander 2013 |
| International Aid | Chinese Finance | AidData 2019 |
| | World Bank | World Bank, AidData 2017 |
| | Global Environment Facility | GEF IEO, AidData 2017 |
| | 18 Country Specific Datasets | Various, AidData |
| Environmental/Geophysical | Land Cover | ESA 2009, MODIS 2013 |
| | Elevation and Slope | NASA 2000 |
| | NDVI | LTDR, NASA 2017 |
| | PM2.5 and Ozone Concentration | Bouma et al. 1997 |
| | Air Temperature | Matsuura and Willmott 2015 |
| | Precipitation | Matsuura and Willmott 2015 |
| Other | Distance to Water | Wessel and Smith 1996 |
| | Distance to Roads | CIESIN and ITOS 2013 |
| | Distance to Country Borders | Hijmans 2015 |

Requested data takes the form of a simple spreadsheet (CSV file) in which each row represents an individual boundary feature (e.g., a city or a district) and each column represents the data selected (e.g., total population in 2000, total population in 2005, or average temperature in 2005). Providing users with data in CSV format allows users without experience working with spatial data, or the computational resources required to process large scale spatial datasets, to still gain the benefits and insights of incorporating subnational spatial data into the analysis and decision-making processes.

## 4. Paving the Way forward for Urban Earth Observations: Collaboration, Communication, and Capacity Development

This paper highlighted some of the opportunities that EO offers to governments in overcoming evidence gaps, and shared various ongoing efforts that can be tapped into by city leaders as well as some of the tools that are already available to improve their decision-making. The paper also suggested ways in which EO can be leveraged for tracking progress and reporting on global sustainable development agendas, which carry urban implications.

As the EO community continues to shift their focus from national level engagements to technical support for the data needs of urban leaders, a similar and matching move is required from cities in embracing geospatial technologies through: (i) active engagement with the EO community and communicating their needs; (ii) building strong collaborative frameworks; and (iii) investing in

developing the necessary capacity within their city government agencies. Meanwhile, national governments need to also support cities in their pursuit of these three activities. Where necessary due to legal or administrative limitations, national or sub-national governments can also take lead in pursuing these three activities on behalf of urban settlements.

To facilitate increased collaboration in the EO sector, GEO invites countries and participating organizations to take part in the GEO Work Program. There are now over 50 activities approved with implementation plans for 2020–2022, and all are open to new membership. Collaborating with GEO under the HPI, EO4SDG or its other work streams will allow city governments and other urban stakeholder organizations to increase their collective impact, build on synergies and scale up best practices that are already working in cities across the globe.

Given the amount of programs already utilizing EO data, it is apparent that better data and information leads to better decision making. However, these benefits are not being effectively communicated to city leaders who are yet to capitalize on the offered opportunities. The EO community would need to improve its communication and advocacy efforts to create the necessary political will to enable this transition. Additionally, the EO community also needs to understand the nuances of local governance and government. City administration varies significantly within and among countries. Therefore, producing standardized data, based on grids or other popular techniques to generate information at scale, might be limiting its use to only higher order governments. AidData's GeoQuery tool has seen reasonable uptake at lower levels of administration possibly because it provides data for administrative boundaries, and in highly accessible spreadsheet format. This may be a valuable insight for the EO community to act on to see higher uptake of EO data at the local level.

Lastly, part of this communication process between the EO community and governments needs to be about the creation of realistic and actionable urban boundaries. Without these boundaries, there will be continued misalignment between the urban data that the EO community generates, using their own definitional criteria, and the data that city leaders would find actionable and the true measure of the status of development in their constituency.

**Author Contributions:** Conceptualization, M.P. and S.R.; methodology, M.P. and S.R.; formal analysis, M.P.; investigation, M.P.; resources, S.R. and A.K.; data curation, M.P. and S.G.; writing—original draft preparation, M.P. and S.G.; writing—review and editing, A.K. and S.R.; visualization, M.P. and S.G.; supervision, M.P. and S.R.; project administration, M.P. and S.R. All authors have read and agreed to the published version of the manuscript.

**Funding:** This research received no external funding.

**Acknowledgments:** The authors would like to thank the following individuals for providing valuable inputs into the development of this paper: Phoebe Odour (RCMRD); Jimena Juarez (INEGI, Mexico); Evangelos Gerasopoulos (National Observatory of Athens); Thomas Kemper (European Commission); Julie Greenwalt (UNEP); Yifang Ban (Royal Institute of Technology, Sweden); Miguel Angel Trejo (INPE, Brazil); Samantha Custer (AidData); and Theadora Mills (GEO).

**Conflicts of Interest:** The authors declare no conflict of interest.

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
