# Peer review of "Open Earth Observations for Sustainable Urban Development"

_remotesensing, doi:10.3390/rs12101646_

Round 1

Reviewer 1 Report

I found this to be an unusually concise and effective manuscript. Let me emphasize that I approach the paper from the standpoint of an urbanist without much technical capacity in this field, so I must leave comments on the data themselves to other reviewers.

The paper discussed the potential of EO data to urban governments. It offers several contextual examples, which are effective.

In terms of what the paper does not do quite so well. First, paragraph 323-33 raised a number of questions. First it was a little hard to follow; ‘paid data’ seems a little clumsy and ‘proprietary data’ might be better. Second, it was not obvious to this reader why such detailed information is significantly more useful than coarser information. The examples presented throughout the paper seemed to link up with fairly broad questions that would not need resolution down to the level of individual streets and homes in order to develop policy. More data = more need for storage and analysis, which might be unnecessarily costly for some potential users. This cost-benefit trade-off needs addressing, perhaps with another example of what users have actually accomplished with the more detailed information.

This leads into a bigger question of increasing salience in the current world. Despite being couched in the context of sustainability, EO remains another leap forward in state surveillance. We already know what can be accomplished by military users who are using satellite imagery to pilot drones and other weapons towards specific buildings. It is not fanciful to imagine that many governments—local, regional—national—will turn away from the most fine-grained data precisely because it is too specific and too powerful.

It should not always be left to the social sciences to raise these kinds of questions. I strongly encourage the authors to discuss issues of privacy and the trade-off between generating data that is usable for most policy situations while maintaining some degree of aggregation. I am also left wondering whether all governments have clean hands with regard to data use. It does not need a great deal of imagination to think about an urban administration plotting the distributions of squatter settlements or homeless encampments so that they can take action to arbitrarily remove them; similarly, with fine grained data it would be possible to classify neighborhoods by religion, based on proximity to churches or mosques, and to allocate infrastructure accordingly. As most cities are socially segregated, this kind of tool could be a sophisticated means for maintaining privilege for a dominant clique within an urban administration. 

This in turn raises another question: why should EO be reserved for governments? It would not be hard to argue that the key intellectual advances in governance throughout much of the world have come from NGOs, who have worked hard to promote social justice concerns in countries where established hierarchies of privilege use the state to control and coerce. This extension of the franchise, so to speak, needs addressing. Would there be a hesitation in directing EO opportunities to non-state actors? And why stop there? There are exciting developments relating to multiple stakeholders in transdisciplinary research, and citizen science could play some role in extending EO into the hands of new users—this would be especially relevant in terms of developing adaptive capacity in urban areas where increased flooding or fire risk is emerging. There, in fact, fine-grained data sets would be valuable.

All in all, an interesting discussion of the technical dimension, but I assert there is more to this issue than the authors have offered us. 

Author Response

Reviewer 1

Date: 14th April 2020

Dear Sir/Madam,

Sub: Changes made to the manuscript and reviewer comments clarifications and response

                First, please allow me to thank you for taking the time to read our manuscript for the review article titled “Open Earth Observations for Sustainable Urban Development.” Your comments and suggestions helped us better understand the impressions generated by the reader and how to fine-tune this messaging further. With this understanding, we have addressed several points that were raised and are comfortable submitting the revised manuscript for your consideration.

                I would like to provide an overview of some changes that we have made to the manuscript to address your comments. Please see below:

  • We have clarified why higher resolution imagery’s transition to the open data domain is important for urban research. You are right in highlighting the fact that many policy decisions can be made using information that can be derived from coarser imagery. However, this paper would like to also acknowledge the importance of high-resolution imagery in helping generate more granular data that can inform specific projects that are undertaken within the umbrella of a broader policy. It is a matter of scale of operation. We did not want to exclude these granular EO data activities from our paper because some of the data that is currently being generated, will eventually reach public domain. You also suggested we include an example of a successful application of high resolution EO data. We think that this will emerge over time if enough visibility around this data’s availability is created. Applications of high-resolution data for sustainable urban development policy are currently very limited, and information is often proprietary. (344-347)
  • The challenge of storage and cost that you mentioned is another good point. We consider this issue to be something that might be best tackled in a separate paper. For the purposes of this paper, the idea of bringing any EO data to the public domain implies that there is no cost of storage to the user. However, challenges associated with its analysis do remain. We acknowledge the issues of local capacities to work with EO data and propose strategic investments into building such capacity in our paper.
  • The implications of EO to issues around government surveillance and use of granular data for potential negative use by the state is beyond the scope of this paper. Such threats are, indeed, not hard to imagine. But this paper is directed strictly towards the use of existing technologies for good in the urban sustainability context. If one looks at the origins of satellites, the technology originated for surveillance. Only through exploring its applications over time in other domains do we see it being used for development.
  • This paper was written from the lens of government decision makers. You are correct in mentioning similar utility to other users of EO data, i.e. advocates, civil society and accountability practitioners. But to include this alternate lens into this paper would dilute the messaging we wish to have. We would love to address the opportunities that EO data (both low- and high-resolution) provides to other group of users in a future paper.

I hope that the above points of clarification and revision have sufficiently improved the quality of our paper. Please do not hesitate to reach out, should further information be required.

Sincerely,

Mihir Prakash

Reviewer 2 Report

General comments

In my opinion, the subject of this review article is relevant for the Remote Sensing journal readers and sufficiently interesting to warrant publication. The research questions are very important. The work sound and the overall approach envisioned and implemented are generally correct. It is evident that the science underlying the discussion is generally good. Figures and tables, generally, are all necessary but with insufficient quality in terms of cartographic and graphic expressions (for the maps).

In summary, I think that this manuscript could be accepted as a review article in the Remote Sensing journal only after some revisions.

Specific comments

Please, see the pdf file in attachment.All comments were performed using the Adobe Acrobat tools - Comment & Markup of Adobe Acrobat DC.

Author Response

Reviewer 2

Date: 14th April 2020

Dear Sir/Madam,

Sub: Changes made to the manuscript and reviewer comments clarifications and response

                First, please allow me to thank you for taking the time to read our manuscript for the review article titled “Open Earth Observations for Sustainable Urban Development.” Your comments and suggestions helped us better understand the impressions generated by the reader and how to fine-tune this messaging further. With this understanding, we have addressed several points that were raised and are comfortable submitting the revised manuscript for your consideration.

                I would like to provide an overview of some changes that we have made to the manuscript to address your comments. Please see below:

  • We have revised the abstract to better reflect the contents of the review article. (9-29)
  • We have adjusted the introduction to better reflect what the authors were trying to achieve. (97-105)
  • We have mentioned the use of some case study examples in the abstract. However, we consider these to be more along the lines of illustrative examples, rather than case studies. (9-29)
  • We have updated two images that were created by the authors to have a north arrow and a scale bar. Images that are sourced from other peer reviewed academic publications, which we have merely reproduced and cited cannot be updated to contain the same cartographic features. (Figure 1 & figure 8)

I hope that the above points of clarification and revision have sufficiently improved the quality of our paper. Please do not hesitate to reach out, should further information be required.

Sincerely,

Mihir Prakash

Reviewer 3 Report

The paper is very well written and when finalised will be a significant contribution to achieving the global sustainable development agenda through promoting evidence-based approaches through earth observation (EO).

However, when I read the paper, I am not sure what is the key goal of the paper because it addresses so many issues but does not go in-depth. What I propose is the writers choose one goal and pursue it to the end, is it to:

  1. offer opportunities to provide timely, spatially disaggregated information
  2. discuss how EO is being and can be leveraged for local decision-making / provide an overview of various efforts that are underway in the EO community to create new tools and databases that can support urban decision-making
  3. highlight untapped data opportunities from EO
  4. highlight barriers that would need to be overcome through better communication and collaboration between the EO community and data users and decision-makers.

Once it is clear, then rewrite the entire paper to address the specific goal.

Author Response

Reviewer 3

Date: 14th April 2020

Dear Sir/Madam,

Sub: Changes made to the manuscript and reviewer comments clarifications and response

                Please allow me to thank you for taking the time to read our manuscript for the review article titled “Open Earth Observations for Sustainable Urban Development.” Your comments and suggestions helped us better understand the impressions generated by the reader and how to fine-tune this messaging further.

First, we really appreciated you highlighting the four distinct themes that emerge from this paper that are necessary and relevant in the context of applications of earth observation to sustainable urban development. However, after a discussion among the authors, we concluded that the objective here was to offer a higher-level discussion that touches upon all four of these points.

Second, we do acknowledge the value of making the second point more salient. However, the extent to which we could cover more use cases from the urban decision-maker’s perspective had to be limited in this paper, in order to maintain the breadth of information that this paper is trying to share. We see expanding on your points (particularly point #2) with a deep-dive approach as opportunities for future articles on these topics.

I hope that you understand the rationale behind our decisions for this review article. Please do not hesitate to reach out, should you require further information.

Sincerely,

Mihir Prakash

Round 2

Reviewer 3 Report

As I mentioned in my first review, the paper has many objectives which weaken the author's ability to influence readers because readers get lost trying to find meaning and structure in the paper and what the authors truly want to convey to the readers.

I understand that they want to have a high-level discussion, however, their explanation is not sufficient to justify having multiple competing objectives within one paper. It may be overwhelming for readers.

When I read the revised abstract the first objective is: to provide a review of ways in which EO data can support measuring various dimensions of sustainable development at the urban scale. If they could stick to this objective and faithfully follow through with this one objective then we will easily understand what they intend to do.

There was nothing much to review since the paper is still the same. However, I noted that the revisions are not of the same quality as the first draft, I think they were written in a hurry - so make sure that you tighten the grammar with the revised parts.

Author Response

Reviewer 3

Date: 28th April 2020

Dear Sir/Madam,

Sub: Changes made to the manuscript and reviewer comments clarifications and response

                Please allow me to thank you for taking the time to read our 2nd version of the manuscript for our review article titled “Open Earth Observations for Sustainable Urban Development.”

Upon revisiting the paper after reading your comments we tried to make the objectives of the paper more salient through introduction of an illustration (Figure 1) as well as reframing the language throughout the manuscript. We have also revised the abstract to more closely reflect the contents of the paper.

We acknowledge that the manuscript largely is the same, but we hope that the modifications we have made to section headings, minor adjustments to various parts of the narrative and clearly setting the stage for the paper through revision of the abstract and addition of a summary graphic, sufficiently address your concerns of overwhelming the readers.

I hope that you understand the rationale behind our decisions for this review article. Please do not hesitate to reach out, should you require further information.

Sincerely,

Mihir Prakash
